# Concordance between US and MRI Two-Dimensional Measurement and Volumetric Segmentation in Fetal Ventriculomegaly

**DOI:** 10.3390/diagnostics13061183

**Published:** 2023-03-20

**Authors:** George Hadjidekov, Gleb Haynatzki, Petya Chaveeva, Miroslav Nikolov, Gabriele Masselli, Andrea Rossi

**Affiliations:** 1Department of Radiology, University Hospital Lozenetz, Koziak 1 Str., 1407 Sofia, Bulgaria; 2Department of Physics, Biophysics and Radiology, Faculty of Medicine, Sofia University “St Kliment Ohridski”, 1504 Sofia, Bulgaria; 3Department of Biostatistics, College of Public Health, University of Nebraska Medical Center, Omaha, NE 68198, USA; 4Department of Fetal Medicine, Shterev Hospital, 1330 Sofia, Bulgaria; 5Department of Theoretical Electrical Engineering, Technical University, 1156 Sofia, Bulgaria; 6Radiology Department, Umberto 1 Hospital Sapienza University, 00161 Rome, Italy; 7Neuroradiology Unit, IRCCS Istituto Giannina Gaslini, Via Gerolamo Gaslini 5, 16147 Genoa, Italy; 8Department of Health Sciences, University of Genoa, 16126 Genoa, Italy

**Keywords:** fetal ventriculomegaly, three-dimensional ultrasonography, 3D-US, virtual organ computer-aided analysis, VOCAL, fetal MRI, fetal ventricular and brain segmentation, 3D-Slicer

## Abstract

We provide a study comparison between two-dimensional measurement and volumetric (3D) segmentation of the lateral ventricles and brain structures in fetuses with isolated and non-isolated ventriculomegaly with 3D virtual organ computer-aided analysis (VOCAL) ultrasonography vs. magnetic resonance imaging (MRI) analyzed with 3D-Slicer software. In this cross-sectional study, 40 fetuses between 20 and 38 gestational weeks with various degrees of ventriculomegaly were included. A total of 71 ventricles were measured with ultrasound (US) and with MRI. A total of 64 sonographic ventricular volumes, 80 ventricular and 40 fetal brain MR volumes were segmented and analyzed using both imaging modalities by three observers. Sizes and volumes of the ventricles and brain parenchyma were independently analyzed by two radiologists, and interobserver correlation of the results with 3D fetal ultrasound data was performed. The semiautomated rotational multiplanar 3D VOCAL technique was performed for ultrasound volumetric measurements. Results were compared to manually extracted ventricular and total brain volumes in 3D-Slicer. Segmentation of fetal brain structures (cerebral and cerebellar hemispheres, brainstem, ventricles) performed independently by two radiologists showed high interobserver agreement. An excellent agreement between VOCAL and MRI volumetric and two-dimensional measurements was established, taking into account the intraclass correlation coefficients (ICC), and a Bland–Altman plot was established. US and MRI are valuable tools for performing fetal brain and ventricular volumetry for clinical prognosis and patient counseling. Our datasets could provide the backbone for further construction of quantitative normative trajectories of fetal intracranial structures and support earlier detection of abnormal brain development and ventriculomegaly, its timing and progression during gestation.

## 1. Introduction

Ventriculomegaly (VM) is a common finding in fetal imaging and a proxy for potential neurological complications. Detected with fetal ultrasound in approximately 1% of all fetuses, this condition is defined as enlargement of the lateral ventricles in utero [1,2]. The frequent association with various central nervous system (CNS) and non-CNS anomalies requires a rigorous prenatal examination with appropriate imaging modalities.

Three-dimensional ultrasound (3DUS) enhances the display of complex anatomical structures traditionally visualized with two-dimensional ultrasound (2DUS), enabling more robust prenatal diagnosis. This is easily achieved by the possibility of the multiplanar format in 3DUS to image an anatomical structure simultaneously in three planes and thus obtain anatomical 3D models of the area of interest. While this tool is an enhancement over 2DUS, it is not currently able to overcome the inherent artifacts and limitations of US. Three Tesla magnetic resonance imaging (3T MRI) was approved for use in humans in 2002, and numerous advantages (e.g., improved tissue resolution and image quality, higher temporal and spatial resolution) have enabled clinicians to implement this modality in routine prenatal practice. Fetal MRI is superior to US and capable of detecting up to 50% additional abnormalities in apparently isolated ventriculomegaly [2]. However, considering cost-efficiency, US is used for screening, and MRI is used as a confirmatory diagnostic tool.

The tremendous transformation of the form and shape of the lateral ventricles during gestation is also associated with the development of the surrounding cranial structures. At early gestational age, the lateral ventricles occupy a proportionally larger volume of the fetal brain. With maturation of the grey matter and folding of the cerebral surfaces, at around 16 weeks of gestation the lateral ventricles decrease in volume and become recognizable on imaging due to formation of four distinct horns. Although the atrial diameter remains constant, by term, the lateral ventricles appear smaller compared to the developed brain [3]. This dynamic ventricular evolution is pivotal in the work-up of fetal ventriculomegaly. Therefore, normal reference values for every gestational week (GW) should be implemented and referenced in clinical practice [4,5].

With this comparative review of two-dimensional versus volumetric measures, and US versus MRI, we aim to bring new insights into the patterns of growth of the lateral ventricles contrasted to the brain parenchyma in isolated and non-isolated forms of ventriculomegaly. We anticipate that these data may improve prenatal counseling, management decisions strategies and pregnancy outcomes. We aim to quantify fetal brain development measured by the degree of ventriculomegaly and to provide a resource for developmental growth trajectories of brain growth in ventriculomegaly. Creation of an atlas-based reference guide from brain parenchymal tissue segmentation in normal and pathological findings, as well as enhancement of existing atlases, could be considered potential applications of our measurements.

## 2. Materials and Methods

This was a cross-sectional study of data collected from selected singleton pregnancies with either isolated or non-isolated ventriculomegaly associated with other CNS anomalies. We excluded multiple pregnancies, a fetus presenting with ventricular wall rupture, cases of schizencephaly, porencephaly and holoprosencephaly, oligohydramnios and pregnancies with contraindications for MR examination such as claustrophobia and mothers with metallic implants. Patients were recruited at University Hospital Lozenetz and Dr Shterev Hospital, Sofia, Bulgaria (1 October 2018 to 30 September 2022) in a collaborative study. All cases were evaluated with initial 2D ultrasonographic anatomical evaluation. Subsequent assessment of the fetal brain with high-resolution 3D US imaging and follow-up fetal MRI were performed within 48 h. All detected fetal brain anomalies were registered and documented in a prenatal database system. Ventriculomegaly was defined as mild (10–12 mm), moderate (12.1–15 mm) and severe (15 mm or more), in accordance with atrial width [6].

Approval from the local Institutional Review Board was obtained for using data in the presented paper, and written informed consent was provided by the parents. Our study group consisted of 40 cases: in 12 cases, the primary diagnosis was made by referral sonographers; in the other 28 cases, diagnosis was made by clinicians who had obtained The Fetal Medicine Foundation Certificate in ultrasound examination for fetal defects during routine visits. Ultrasonography data revealed ventriculomegaly ranging between 10 mm and 30 mm in 71 ventricles. All participants were offered an additional clinical fetal MRI for correlation and assessment of additional pathologies. In the third trimester (after 31 weeks), 3D US measurements were performed on a single ventricle in 15 cases, due to reverberation artifacts from the head positioning in the maternal pelvis, precluding a precise volumetric analysis.

All obtained MR study scans achieved the quality and acquisition criteria for image analysis. Therefore, 71 ventricular sizes, 71 (with VM) from a total of 80 measured ventricular volumes and 40 brain volumes, were analyzed with MRI.

### 2.1. US Technique and Protocol

Two-dimensional ultrasound examinations were performed with Voluson E10 system (GE Healthcare, Kretz Ultrasound, Zipf, Austria) using a 4 to 8 MHz transabdominal probe. Quantitative 2D measurements were recorded for right and left ventricles for each case, in conjunction with that for the fetal position relative to the maternal abdominal wall. This was used to prevent misinterpretation, equivocal measurements and possible errors in comparisons between methods. Lateral ventricle (LV) size was assessed by the measurement of the atrial width in the axial transventricular plane at the level of the parieto-occipital fissure [7]. For volumetric assessment with 3D, we used 4D View Version 6 software and virtual organ computed-aided analysis (VOCAL). Multiplanar volume calculation was based on the manual area delineation of each image included in the volume of the three-dimensional rendering planes. A wide sweep angle higher than 60° was applied for volume acquisition of the total fetal head in the 3D dataset. The total fetal brain volume was calculated using the fetal BPD plane as a landmark, including the falx cerebri, the cavum septi pellucidi and thalamic nuclei. In a similar way, we used VOCAL mode 30° to calculate more irregular ventricular volumes. The three-horn view plan was used as a landmark, as previously described and validated in the literature, for each ventricle. The selected image resolution varied from high to maximum. For optimal volumetric measurements by VOCAL mode in concomitant visualization of the three-horn view plan, the ventricle was adjusted in a position of its anteroposterior axis to be perpendicular to the vertical rotation axis [8,9].

### 2.2. MRI Technique and Protocol

All patients were examined in supine position with a pillow under the knees without the need for a left lateral decubitus position to prevent inferior vena cava compression syndrome. To reduce claustrophobia, patients were scanned feet first with the uterus centered in the bore. No sedation was required. Patients were scanned using 1.5T Signa (GE Healthcare, Chicago, IL, USA) and 3T Ingenia (Philips Healthcare, Amsterdam, The Netherlands) superconducting systems. A 32-channel surface coil was wrapped over the abdomen of the patient in order to place the uterus between the tabletop and the surface coil while using 3T MRI. In 1.5T, an 8-element phased-array surface coil was positioned over the abdomen. The entire uterus was localized with a large field of view covering the cervical length and the entire placenta. Images of the entire uterus in axial, sagittal and coronal planes were captured using T2-weighted half-Fourier single-shot fast spin echo (ssFSE) and turbo spin echo (ssTSE), with large field-of-view and slice thickness ranging between 3 mm and 4 mm, enabling the determination of fetal situs and right-to-left side orientation. T1 sequences were routinely used to exclude hemorrhage or meconium content and were not used for volumetric postprocessing purposes. In selected cases, additional advanced MRI techniques such as echoplanar imaging (EPI) and diffusion-weighted imaging (DWI) sequences were included in the standard protocol to rule out concomitant pathology, but not for the volumetric and segmentation analysis. We used ssFSE in 1.5T and ssTSE in 3T sequences in three planes for the fetus, with the following parameters: field-of-view from 24 × 18 for the fetal head up to 40 × 40 cm for the entire fetus; matrix 512 × 256, 296 × 246, 256 × 256; slice thickness 3 mm and 4 mm; intersection gap 0 mm; single-shot sequence acquisition time 29–56 s for 25–40 slices; number of excitations (NEX) −1 and flip-angle 70°. The TR/TE parameters were 2300/70 ms in 3T and 1360/89 ms in 1.5T. The scanning parameters correspond to the routine scanning protocol in clinical use, and these were not adjusted for the purposes of volumetry in the present study. All the sequences were performed with no breath-holding, assuring optimal maternal comfort during imaging. The overall duration of the scanning time using ssFSE and ssTSE sequences ranged from 6 to 20 min in both 1.5T and 3T, including localizer and repeated sequences due to fetal motion or adjustment for better anatomical orientation. All MR studies were performed by an experienced technologist trained in fetal MRI, and all images were reviewed simultaneously by an experienced radiologist before the end of the exam in order to further reduce the total exposure time. After completion of the exam, a radiologist with 10 years of experience in fetal MRI together with a resident performed independent and double-blinded standardized bi-dimensional and volumetric measurements of the lateral ventricle sizes at the level of the atrium where both third ventricle and posterior basal ganglia were visible, perpendicular to the ventricular long axis. Both observers classified the degree of dilation (mild, moderate, severe) and reported additional (if any) CNS abnormalities (Figure 1).

### 2.3. MRI Safety

In all pregnant patients, the benefit of performing fetal MRI over the potential risk was evaluated by an experienced radiologist. Radiofrequency fields generate heat and are measured by specific absorption rate (SAR). The FDA safety limits are set at 4 W/kg per fetal examination for every magnetic field strength. Modern MR units, including 3T imagers, have build-up failsafe controls disallowing additional overlimit exposure. Similar to all communicated reports, our SAR values were far below exposure limits for both 1.5T and 3T [10]. Another possible safety concern is the average effective exposure time during fetal MRI study. However, exposure time is not continuous during examination and accounts for only about 33% of the total study time; the remaining time is reserved for pre-positioning of the coil and planning sequences [11]. We assume that performing fetal scans by a trained radiographer together with the radiologist could further shorten the overall duration of the examination time. With the growing availability of 3T and higher Tesla magnets, many centers are beginning to address the use of 3T imagers for fetal examinations. Despite the inherent system SAR limitations, radiographers should strictly follow the established guidelines for SAR limits and consider the possibility of higher power deposition using high-field imagers [10,12,13,14].

### 2.4. Comparison of US and MRI Measurements

Both interobserver (images reviewed by experienced fetal ultrasonographer and radiologist) and intraobserver (lateral ventricular sizes and volumes, as well as total brain volumes, measured and segmented manually, independently by experienced and non-experienced radiologists) comparisons between ventricular and total brain parenchymal measurements were assessed. We chose the MRI measurement of the ventricle as a default parameter for the degree of ventricular enlargement (10–12 mm, 12.1–15 mm, and more than 15 mm) to stratify fetuses into these groups.

### 2.5. Volume Segmentation and Quantification of 3D Analysis

While linear sizes were measured on an IPS Philips Healthcare workstation, volumetric measurements and reconstructions were performed using 3D-Slicer platform version 4.11. We also used the following software for MR imaging processing: SC PACS Server, Version 2.3.0.0 for archiving selected studies; SC PACS Viewer, version 2.5.0.1 for anonymization of patient data. We created a local DICOM database [Google Drive] containing 40 fetal MR studies in the SC PACS Server. A radiologist experienced in fetal MRI selected the most suitable axial series with slice thickness of 3 or 4 mm for each study. The processed anonymized studies were imported into the software product 3D-Slicer for manual segmentation and volume measurements of the fetal supratentorial brain, brainstem, cerebellar hemispheres and ventricles, as well as calculations of the total ventricular volume, total brain volume and brain volume without ventricles. 3D-Slicer is an open-source software for the analysis and display of DICOM imaging datasets and for scientific visualization of 3D reconstructions [15,16]. Volume segmentations obtained with 3D-Slicer and VOCAL are illustrated in Figure 2.

Once the selected series were uploaded in 3D-Slicer in the axial slice window, we added four different layers, entitled “cerebrum”, “stem”, “cerebellum” and “ventricle”, and carried out processing from various modules, e.g., Segment Editor, Segmentation, Master Volume. The listed brain structures were outlined, and their contours were manually delineated in 12 to 16 axial planes (slices) depending on the size of the structure and the slice thickness, to cover the entire volume with successive axial slices in superior to inferior direction using the “Draw tool”. The manual hand-tracing of all consecutive slices in segmentations was achieved using a drawing tablet (Microsoft Surface Go2 Tablet). The cerebrum was defined as the supratentorial brain parenchyma above the tentorium, excluding the brainstem, the cerebellum and the ventricles. The total ventricular volume was designated as the volume of both right and left lateral ventricles, excluding the third and fourth ventricles and cavum septum pellucidum (CSP). The choroid plexus was included in the respective ventricular volume. The stem volume includes all brainstem structures, while the cerebellar volume was attributed to the volume of both cerebellar hemispheres and vermis, excluding the fourth ventricle. The total brain volumes were obtained by summing the volumes of the supratentorial brain, brainstem, cerebellar hemispheres and ventricles, while the sum of the mentioned volumes, excluding the ventricular volume, was calculated as total brain volume without the ventricles. Volumetric segmentations of fetal brain and ventricles were performed at different gestational ages (Figure 3A–C) and in selected projections (Figure 3D,E) with 3D-Slicer and with VOCAL (Figure 3F).

### 2.6. Statistical Analysis

Continuous characteristics are presented as medians (interquartile range, IQR: Q1, Q3) since these are more stable than means. Categorical characteristics are presented as counts and percentages (*n*, %). Correlation between continuous variables was assessed with the nonparametric Spearman’s rank correlation coefficient. Since more than one rater was involved in the study, intra- and inter-rater reliability were performed for all measurements. Intra-rater and inter-rater variability were assessed using the intraclass correlation coefficient and Bland–Altman plots. The intra- and interclass correlation coefficients for all 2D and 3D measurements were 0.85 (*p* < 0.01). The average relative growth rate for each structure represents the percent volume change relative to the median volume over the whole range of the time variable (here, gestational age in weeks, GW). This was calculated assuming linear growth to make our results comparable to previous studies. Robust (linear) regression was used in order to detect outliers and provide resistant (stable) results in the presence of outliers [17,18]. The level of significance was alpha = 0.05. All statistical analyses were performed using the statistical software SAS version 9.4 (SAS Inst. Inc., Cary, NC, USA).

## 3. Results

Forty women with singleton pregnancies and various degrees of VM were included in the study. Mean maternal age was 30 years (ranging from 22 to 41 years), and mean gestational age was 29 weeks (ranging from 20 to 38 GW). Seventy-one (71) ventricle sizes were measured with US and MRI due to nine cases of unilateral VM. Sixty-four total ventricular volumes (35 left and 29 right) were measured with US. The remaining volumes were excluded from analysis due to US reverberation artifacts. All 80 ventricle volumes were manually measured with MRI, with 71 of them (with VM) used for comparative analysis. Ventricular atrial diameter was manually measured on axial MRI. The range of the diameter was 10.2 to 36.8 mm (average: 14.17 mm) in one or both hemispheres (average of 14.46 mm for the left ventricle and average of 13.87 mm for the right). The degree of VM was established as mild (10–12 mm) in 39.5% (*n* = 29), as moderate (12.1–15 mm) in 38% (*n* = 27), and as severe (>15 mm) in 22.5% (*n* = 16). Isolated forms of VM were observed in 22 fetuses (55%), and of those, 17 were bilateral (42.5%), 3 were left-sided (7.5%) and 2 were right-sided (5%). In the remaining 18 fetuses (45%), additional anomalies were identified. Prenatal mortality was 7.5% (3/40). The ventricular two-dimensional and volumetric median measurements are presented in Table 1.

Our observations from the 3D volumetric measurements showed a moderate increase in the volume of the lateral ventricles with increasing GA compared to an exponential increase in the volumes of cerebral, brainstem and cerebellar structures. Moreover, a discrepancy between relative growth rates in the two different sub-groups of lateral ventricles was noted—the growth pattern of the ventricles in cases with additional pathologies was much larger compared to the ventricular volume growth in isolated VM. A detailed presentation of 3D MRI biometric fetal brain calculations is provided in Table 2.

Presentation of the same trends can be observed in Figure 4, Figure 5, Figure 6, Figure 7 and Figure 8. The figures show variables of growth trajectories based on the volumetric MR calculations of cerebrum, total ventricle volume, stem, cerebellum and total brain volume in isolated and non-isolated cases of VM. Figure 4 demonstrates the growth of brain parenchyma in mm^3^ with increase in GA, which did not significantly differ by group (isolated vs. non-isolated). The growth of the lateral ventricles was greater overall in the non-isolated group over the entire GA range (20–38 GW) (Figure 5). Growth of the brainstem (Figure 6), the cerebellum (Figure 7) and the total brain (Figure 8) showed no significant differences between the two groups with the increase in GA.

### 3.1. Detection of Additional Anomalies with US and MRI

Associated CNS abnormalities in fetal VM are common. In severe VM, additional CNS anomalies are seen in 50–85% of cases. In all 22 isolated cases of ultrasonography-detected VM, MRI was confirmatory and in agreement with the ultrasound (100%). In the remaining 18 fetuses (10 with severe, 5 with moderate and 3 with mild VM), VM and additional CNS anomalies were observed, including agenesis of the corpus callosum (*n* = 8), corpus callosum hypoplasia and dysplasia (*n* = 4), cerebral ependymal and germinal matrix intraventricular hemorrhages (*n* = 3), polymicrogyria (*n* = 2), choroid plexus papilloma (*n* = 1), microcephaly (*n* = 1) and lissencephaly (*n* = 1). Referral sonographic diagnosis agreed with MR only in 70% (28/40) and neurosonography in 82.5% (33/40) of the cases. Neurosonographic diagnosis agreed with MR in 97.5% of the cases (39/40). In all cases, we compared the diagnosis of the referral sonographers with the diagnosis of the expert in neurosonography and, finally, with the MRI diagnosis. For example, in one case of microcephaly, MRI was in agreement with the US diagnosis of both referral sonographic and neurosonography examinations, where in one case of suspicion of partial callosal agenesia, MRI did not confirm either of the US diagnoses proposed by the referral sonographer or the expert in neurosonography. MRI was able to diagnose additional malformations to referral ultrasound mostly in cases with multiple cerebral anomalies or in cases with choroid plexus papilloma and polymicrogyria.

### 3.2. Correlation between 3D Volumes Measured with US and MRI

A strong correlation was detected between US and MRI measurements for left ventricle volumes (ICC = 0.98), right ventricle volumes (ICC = 0.98), total ventricle volumes (ICC = 0.99) and total brain volumes (ICC = 0.98). Similarly strong correlation was observed for left ventricle sizes (ICC = 0.96) and right ventricle sizes (ICC = 0.96). High interobserver agreement in volumetric measurements of different supra- and infratentorial fetal brain segments was also observed (ICC = 0.85, *p* < 0.01).

### 3.3. Time for Postprocessing Segmentation

While for the 3D-US VOCAL measurements and the semi-automated segmentation the time for processing varied from 5 to 10 min, the manual MRI postprocessing using 3D-Slicer was a more time-consuming procedure. Manual segmentation offers many advantages over automated segmentation: it tends to be more accurate and, in our experience, provides more realistic volumetric models. However, significant effort is required to obtain detailed and accurate manual segmentation. In our practice, the process to perform complete brain segmentation of a single fetus ranges from 45 to 50 min at the beginning, decreasing to 25 min after experience is gained. In automatic segmentation techniques, substantial manual editing is often required, especially in the context of even minor fetal movement artifacts, meaning that the total postprocessing time can be similar to that for manual segmentation. No re-editing of the segmentation for uncorrected or blurred area removal was necessary in our calculations.

## 4. Discussion

The lateral ventricles become visible at 13–14 GW, and they change considerably in size and shape during cerebral development. Prominent occipital horns could be a normal finding until 24 GW [19]. Between 24 and 28 weeks they look less protuberant and appear small at 34–36 weeks [20]. Despite the evident continuous change in the form and shape, the atrial diameter of the lateral ventricles remains relatively constant from 14 to 40 weeks’ gestation, measuring less than 10 mm in normal fetuses [21,22]. With the same cutoff value for VM of more than 10 mm, fetal MRI and US can produce inconsistent results, with US providing measurements of atrium width at the level of the thalamic nuclei that are from 1 to 2 mm larger [23]. Both fetal ultrasonography and MRI have been used for measurements of intracranial structures and volumes of the ventricles and brain parenchyma in utero across the gestational period to derive and contribute normative growth trajectories [24]. Some studies have aimed to quantify and establish a reference of intracranial structure volumes in normal fetuses and to create a normative database [25,26,27]. Other studies have also attempted to establish a normative MR dataset for intracranial fetal volume at different gestational ages using non-reconstructed data, sensitive to fetal motion [28,29]. The broad range of standard deviations (SD) and standard errors of the mean (SEM) in different studies makes it difficult to confidently diagnose true pathology. Thus, volumetric measurements and results should be carefully assessed and analyzed. For example, some of the cited MR volumetric studies have communicated contradictory results regarding whether the total ventricular volume increases [30], decreases [28] or undergoes minor changes with progression of gestational age [24]. The broad variations in these reports are more likely to be due to methodological errors, rather than biological variations. The variability of data may be due to differences in the anatomical delineation of the structures, such as including the cavum septum pellucidum (CSP) or the third ventricle into the total lateral ventricular volume. Therefore, it will be highly advisable to establish guidelines for further volumetric studies in order to standardize the measurements and obtain unequivocal, reliable and non-contradictory results. Such studies, including our study, can be beneficial in the understanding of the progression and timing of normal fetal brain development, brain aberrations and early prenatal detection of deviation from normal growth. Based on the largest cohort of 659 examined normal fetuses, Shi et al. established a method for automated fetal brain analysis together with brain extraction, 3D volumetric reconstruction, atlas generation and brain development quantification [31]. This approach decreases the time necessary for manual segmentation. However, time-consuming manual extraction is more accurate and provides realistic values with close-to-anatomical final 3D software segmented images. Moreover, the time required for manual segmentation is reduced after gaining experience with the software and is comparable with manual editing following automated segmentation. We found 3D-Slicer to be a user-friendly platform, providing reliable brain and ventricular volumetric reconstructions with realistic volumes, comparable in values to those from 3DUS. The full analysis could be achieved in less than 30 min, which is approximately equal to the time it takes when using manufacturer software with manual re-editing for each patient. We assume that study data using precise manual segmentations, as in our case, could be valuable for future design, verification and comparison of automatic volumetric segmentation methods. Contributing our 2D and 3D fetal brain biometric parameters to other study open access databases could help to expand existing datasets and to develop new growth trajectories. The compilation of our and similar data could improve understanding of brain growth in normal fetuses and in pathological deviations from normal development in high-risk pregnancies or premature deliveries [32].

As a valuable adjunct to US, fetal MRI benefits from better evaluation of subtle intracranial abnormalities such as parenchymal damage, hemorrhages or cortical migration disorders [33,34,35]. A recent meta-analysis by Di Mascio et al. shows lower than previously reported rates of CNS anomalies exclusively detected with MRI following dedicated neurosonography. A major conclusion was the excellent diagnostic performance of MRI in the third trimester for identification of additional CNS anomalies. The results from their study are identical to our conclusions, describing better detection rates of intracranial hemorrhages and some cortical and white matter anomalies [36]. It is known that the occipital horns demonstrate typical internal and medial deviation in the early third trimester, due to the evolution of the calcar avis [19]. Some studies demonstrated that fetuses with isolated mild VM have larger lateral ventricular volumes and smaller calcarine sulcus depths [37]. Our data show similar findings to other studies regarding the pattern of growth of fetal brain structures and ventricles, for example, as reported by Scott et al. [26]. While cerebral, brainstem and cerebellar segments showed an exponential increase with advance of GA in both isolated and non-isolated groups, the volume of the lateral ventricles increased moderately. Moreover, the average growth rate was significantly different in isolated (3.47%) and non-isolated (5.48%) cases of VM. We observed a pronounced acceleration of the volumetric growth pattern of the ventricles in cases of additional pathologies compared to the ventricular volume growth in isolated VM. Our results revealed that the average growth rates of the volumes varied between structures, with fastest growth of the cerebellar hemispheres (22.32%), followed by supratentorial cerebral hemispheres (9.86%) and brainstem (7.76%), in contrast to the slowest growth of the lateral ventricles. In summary, the average growth rate per week for the total brain (10.06%) was more than double that of the dilated lateral ventricles (4.58%). In our cohort, none of the cases diagnosed with isolated ventriculomegaly with MRI proceeded into pregnancy interruption. The additional information obtained with MRI in this cohort group did not alter the parental decisions towards the continuance of the pregnancy and the delivery options. In the non-isolated group, three women (or 7.5% of cases) underwent termination of pregnancy. The added value of fetal MRI to US in cases of VM has been reported in several studies [33,34,38]; however, ultrasound, as an inexpensive, widely available and time-efficient technique, has an advantage over MRI to monitor ventricular size once the diagnosis of VM is established [39]. Our observations have shown that the lowest incidence of additional abnormalities with MRI after ultrasonography is seen in cases of mild ventriculomegaly. Similar to previous studies, the risk of identifying additional pathologies is lower in isolated, compared to non-isolated, cases [40].

Computerized analysis is an important module contributing to reproducibility and efficiency of quantitative imaging techniques for clinical research applications. Similar to Radiology Workstation, a robust tool such as 3D-Slicer automatically performs quantification and segmental statistics, with excellent visualization capabilities and a user-friendly interface. It provides a clear presentation of the calculated results from any desired structure segmentation, including volume, surface area, mean intensity, scalar volumes and various other metrics for each segment in mm^3^ and cm^3^ [41].

With the expanding availability of 3T scanners, fetal studies benefit from the improvements in the technology. Many limitations with respect to image quality, such as fetal motion, will be corrected with recent advances in image acquisition and postprocessing techniques [42]. The greater SNR allows for implementation of parallel imaging to further speed up the ssTSE protocols [10,13,43]. Faster and robust MRI sequences minimizing or freezing fetal motion are constantly evolving, as are the postprocessing methods for reconstructing 3D representations of the fetal brain. These improvements can boost the availability of normative value datasets for fetal brain structures and, in this way, facilitate the standardization of different volumetric measurements and segmentation algorithms. The implementation of fetal biometrics into clinical practice requires continuous ongoing validation and may offer advanced fetal diagnostic capabilities for accurate identification and monitoring of the compromised fetus.

### Limitations and Future Perspectives

A limitation of our study stems from the limited number of cases due to the exclusion criteria, including non-isolated VM with other-than-CNS anomalies. An increase in the number of fetal brains will assure quantitative stability of our findings and more robust statistical analysis results and will enable the acquisition of expert skills and performance in the process of manual segmentation. A promising extension of our study would be assessment of residual parenchymal volume (without ventricles) by calculation of the ratio of ventricular volume to total brain volume, which could prove clinically useful for long-term prognosis of fetal brain development.

## 5. Conclusions

The presented normative datasets across different gestational ages could provide a backbone for further establishment of quantitative normative trajectories of fetal cranial structures. Thus, earlier detection of abnormal brain development and ventriculomegaly, its timing and progression during gestation, as well as prenatal counseling and patient outcome could be significantly improved. Despite the promise of new models that incorporate 3D technology and the potential for volume assessment, further studies are necessary to evaluate clinical utility and standardization, prior to implementation into routine practice. Volumetric measurements of the fetal ventricles and total brain volume are feasible, reliable and provide high agreement when using 3D ultrasound VOCAL and fetal MRI 3D-Slicer software techniques. Although it is more time-consuming, manual fetal MR segmentation and volumetric measurements provide more realistic 3D models of brain structures, with excellent intraobserver agreement. Parenchymal and ventricular volumetry and models could be helpful for calculating anatomical development and growth patterns over gestation, estimating postnatal findings, long-term outcomes and prognosis. With technical improvement and expert use, these impressive imaging modalities will further evolve and enhance care for the fetus.

## Figures and Tables

**Figure 1 diagnostics-13-01183-f001:**
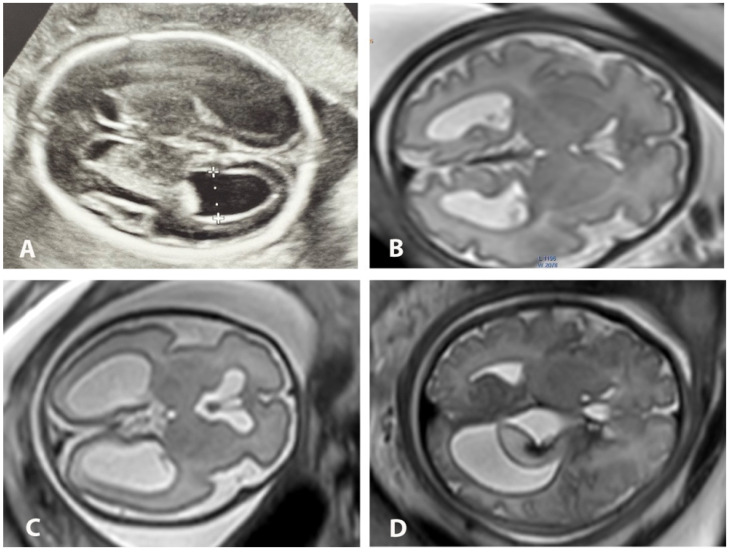
Axial US and MRI images of various degrees of ventriculomegaly. Ultrasonographic image at 28 GW with reverberation artifacts (**A**). Axial MR images of fetal brain with mild bilateral VM at 31 GW (**B**), moderate bilateral VM at 29 GW (**C**) and severe unilateral VM with choroid plexus papilloma hemorrhage (**D**).

**Figure 2 diagnostics-13-01183-f002:**
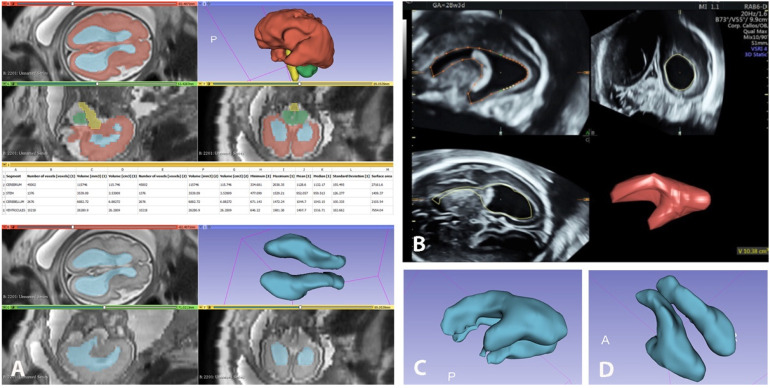
Computed volume segmentations with 3D-Slicer and VOCAL. Fetal brain and ventricle volume calculations using 3D-Slicer (**A**). Ventricular volume segmentation using 3D US VOCAL technique (**B**). Segmented ventricular volumetric reconstructions in lateral and oblique projection of moderate VM (**C**,**D**).

**Figure 3 diagnostics-13-01183-f003:**
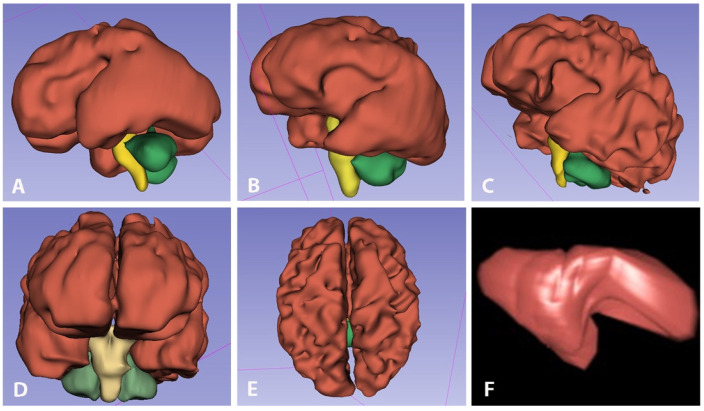
Fetal brain volumetric segmentations with 3D-Slicer. Sagittal views at 26 GW (**A**), 29 GW (**B**) and 33 GW (**C**). Interactive 3D visualization of cerebral hemispheres (red), brainstem (yellow) and cerebellum (green) in frontal (**D**) and craniocaudal (**E**) view. Three-dimensional ventricular volume rendering of moderate VM with VOCAL at 29 GW (**F**).

**Figure 4 diagnostics-13-01183-f004:**
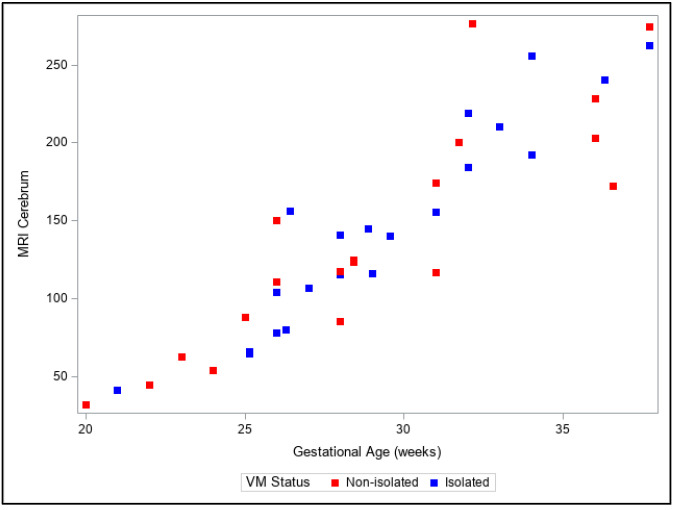
Three-dimensional growth trajectory measurements of supratentorial fetal brain volume in isolated and non-isolated VM.

**Figure 5 diagnostics-13-01183-f005:**
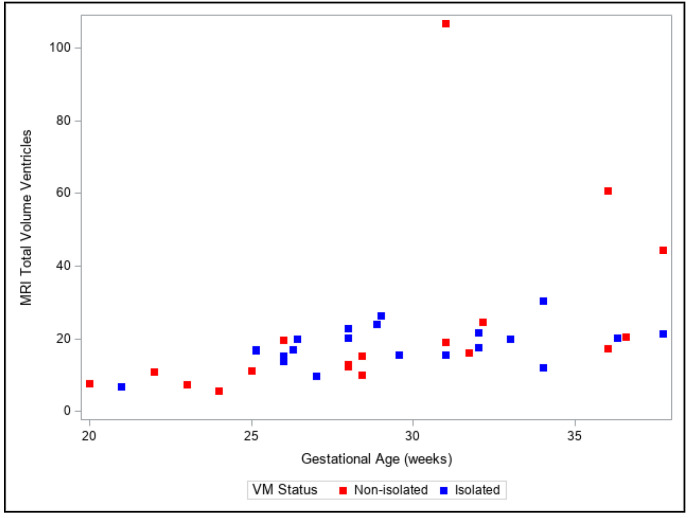
Three-dimensional growth trajectory measurements of ventricle volume in isolated and non-isolated VM.

**Figure 6 diagnostics-13-01183-f006:**
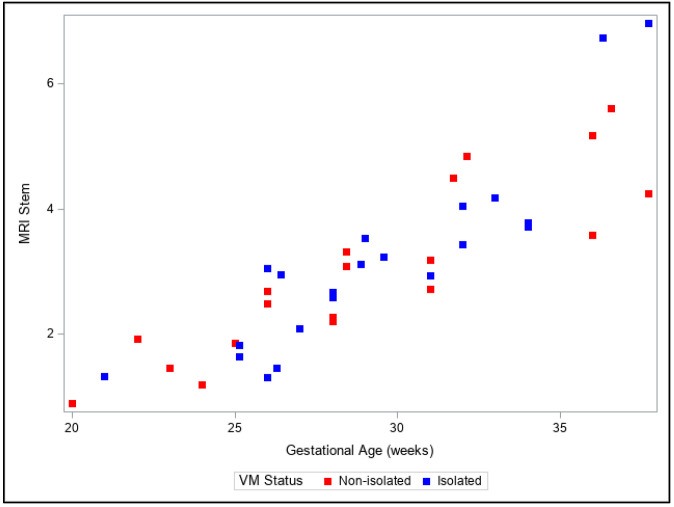
Three-dimensional growth trajectory measurements of brainstem volume in isolated and non-isolated VM.

**Figure 7 diagnostics-13-01183-f007:**
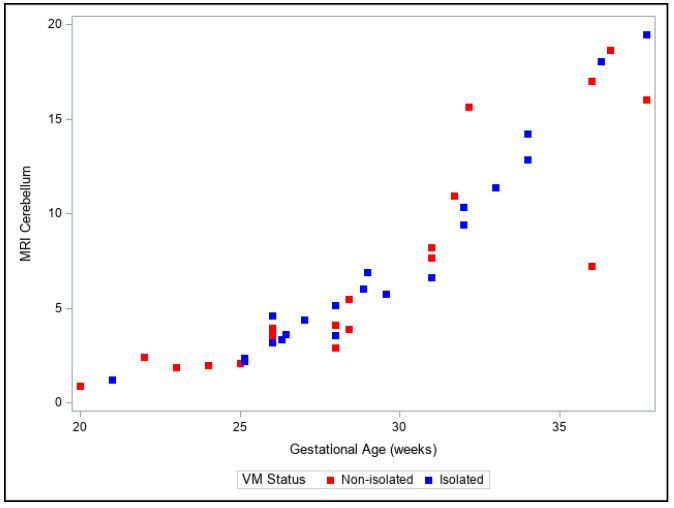
Three-dimensional growth trajectory measurements of cerebellar hemisphere volume in isolated and non-isolated VM.

**Figure 8 diagnostics-13-01183-f008:**
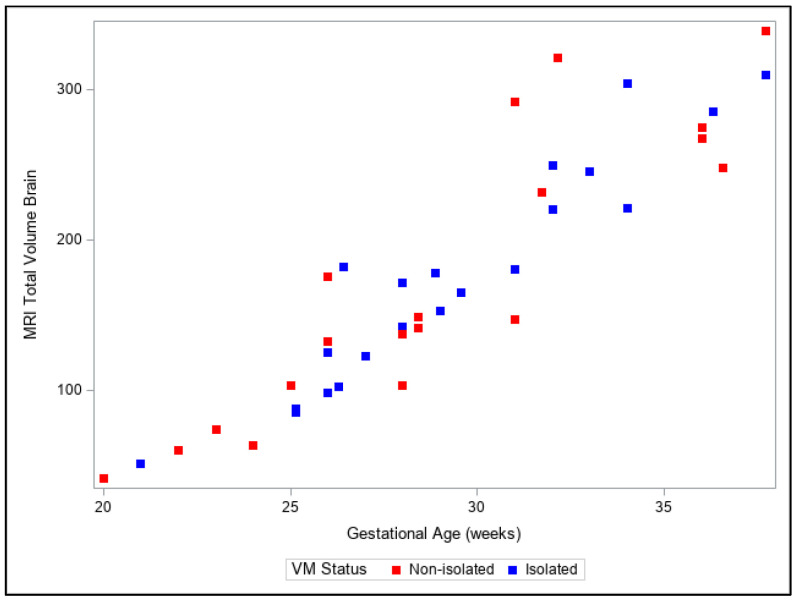
Three-dimensional growth trajectory measurements of total brain volume in isolated and non-isolated VM.

**Table 1 diagnostics-13-01183-t001:** US and MRI ventricular two-dimensional and volumetric median measurements. Abbreviations: VENTR—ventricular.

Ventricular Sizes and Volumes (Medians)		Ventr Size (mm)						Ventr Volume (cm^3^)				
		*n*	US		*n*	MRI		*n*	US		*n*	MRI
ISOLATED VM		37			37			31			37	
mild (10–12 mm)		15	10.90		15	11.50		12	7.71		15	8.36
moderate (12.1–15 mm)		13	13.20		13	13.60		13	9.24		13	9.94
severe (>15 mm)		9	15.30		9	16.60		6	10.21		9	11.32
NON-ISOLATED VM		34			34			28			34	
mild (10–12 mm)		13	11.00		13	11.30		12	3.98		13	5.21
moderate (12.1–15 mm)		14	12.75		14	13.95		11	6.06		14	7.27
severe (>15 mm)		7	25.00		7	25.00		5	27.80		7	29.34

**Table 2 diagnostics-13-01183-t002:** Three-dimensional MRI biometric fetal brain calculations.

	Cerebrum			Lateral Ventricles			Brainstem			Cerebellum			Total Brain Volume		
	All	Non-Isolated	Isolated	All	Non-Isolated	Isolated	All	Non-Isolated	Isolated	All	Non-Isolated	Isolated	All	Non-Isolated	Isolated
Spearman’s r	0.90			0.41			0.88			0.91			0.92		
*p*-value	<0.0001			0.0093			<0.0001			<0.0001			<0.0001		
Average volume (cm^3^)															
At 22 GW	58.40	54.13	64.54	9.18	7.50	16.59	1.55	1.46	1.64	2.01	1.95	2.21	68.12	62.81	84.98
At 26 GW	106.47	130.44	103.73	15.14	17.27	15.14	2.48	2.58	2.09	3.59	3.74	3.59	125.11	154.02	122.72
At 30 GW	123.71	120.60	140.03	15.63	12.57	22.86	3.09	2.68	3.12	5.16	4.01	5.72	148.68	138.88	164.62
At 34 GW	196.35	187.39	201.21	19.35	21.69	18.67	3.75	3.85	3.75	10.64	9.56	10.87	238.60	261.96	233.23
At 38 GW	234.23	215.59	251.29	20.94	32.49	20.70	4.71	3.92	6.85	16.48	11.73	18.77	279.89	271.04	297.60
Average Relative growth rate (%/week)	9.86	9.44	10.87	4.58	5.48	3.47	7.76	8.56	7.56	22.32	23.66	23.16	10.06	10.30	9.75
Absolute growth rate (cm^3^/week)															
22-to-26 GW	48.07	76.31	39.19	5.97	9.77	−1.45	0.93	1.12	0.45	1.59	1.79	1.38	56.99	91.21	37.74
26-to-30 GW	17.24	−9.84	36.30	0.49	−4.70	7.72	0.61	0.10	1.03	1.57	0.27	2.13	23.57	−15.14	41.90
30-to-34 GW	72.64	66.79	61.18	3.72	9.12	−4.19	0.66	1.17	0.63	5.48	5.55	5.15	89.92	123.08	68.61
34-to-38 GW	37.88	28.20	50.08	1.59	10.80	2.03	0.96	0.07	3.10	5.84	2.17	7.90	41.29	9.09	64.37

## Data Availability

The data supporting the findings and the reported results, including analyzed archived datasets, are available from the corresponding author upon reasonable request.

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
