# Peer review of "Concordance between US and MRI Two-Dimensional Measurement and Volumetric Segmentation in Fetal Ventriculomegaly"

_diagnostics, 2023, doi:10.3390/diagnostics13061183_

Round 1
Reviewer 1 Report
The manuscript proposed for publication by
Hadjidekov et al is interesting but needs an English revision .
The results are well presented, and they are supporting the conclusion. Maybe the authors should add 2 more sections( Limitations and Future Perspectives) . These sections would bring more value to the paper.
Author Response
We are grateful for the thorough review and we have addressed all the comments by the Reviewer. We are very glad to see that the Reviewer considers the study interesting and that the results are well presented. We appreciate the thoughtful and constructive feedback provided in your review. We have carefully considered your comments and have made the following revisions to the manuscript:
1.With respect to the English Language, we agree and the manuscript has been through an extensive English language revision by a native English-speaking colleague with a substantial experience of publishing numerous articles. We are confident that the typographic, linguistic and grammatical errors have now been diligently corrected. This can be seen by the numerous track changes throughout the word document with respect correcting the English.
2.The Reviewer also suggested that we should add two more sections to the article (Limitations and Future Perspectives). This information was already in the Discussion but we agree that it was not clear as it was not presented as a separate sub-section. Now we have addressed this and have highlighted the corresponding Limitations and Future Perspective sections and have added the appropriate sub-heading.
We believe that you find these changes satisfactory and the revisions have substantially improved the quality of the manuscript and have addressed the issues raised by the reviewer.
Reviewer 2 Report
This is an interesting, well-written, original research article comparing two-dimensional and volumetric measurеments of the lateral ventricles in fetuses with ventriculomegaly by ultrasonography vs MRI. Literature review is adequate and statistical analysis is valid. Conclusions are supported by the results. The study offers the knowledge that US and MRI are equally valuable tools for performing fetal brain ventricular volumetry to help with clinical prognosis and patient counseling. Authors' datasets could serve for further construction of quantitative normative trajectories of fetal intracranial structures. I would be happy for the manuscript to be accepted for publication in its present form.
Author Response
We thank the Reviewer for the time and effort in reviewing our manuscript. We are pleased that the Reviewer is satisfied that the study we have described in the paper has been well executed, interesting and original with adequate references and valid statistical analysis. We appreciate the comment from the Reviewer that the conclusions are supported by the results of our study. We are happy that the Reviewer fully supports the publication of the manuscript.
Reviewer 3 Report
The author provide a comparative review between two-dimensional measurеments and volumetric (3D) segmentation of the lateral ventricles and brain structures in fetuses with isolated or not ventriculomegaly by 3D virtual organ computer-aided analysis ultrasonography VOCAL vs. MRI analyzed by 3D-Slicer software.
The article is well written and explained. The discussion is well written, and the results reflect the purpose of the study. The methods describe precisely how the study was conducted.
Author Response
We thank the Reviewer for the time and effort to review our manuscript. We are pleased that the Reviewer is satisfied that the study we have described in the paper has been appropriately designed, well written, explained and presented with clearly outlined methods and results, and a balanced discussion.
Reviewer 4 Report
Dear Authors
I read with interest your article on comparison of 3D ultrasound and MRI measurements on the fetal ventricular system. In the article you have shown the compatibility of the results in the two modalities; in addition you have shown differences in patterns of brain growth in isolated and non-isolated ventriculomegaly, concluding a different growth trajectories for the two groups.
While such comparisons have been done before, the fact that that the fetal brains with additional structural abnormalities show wider divergence in growth in late pregnancy comparing to the cases of isolated ventriculomegaly is a significant finding. However while the isolated ventriculomegaly group is a relatively uniform group, I am not sure calling this divergence a growth of non-isolated ventriculomegaly a growth trajectory is proper in a very heterogeneous group.
I also have some isolated suggestions, and find some issues in the overall presentation of the material.
First the isolated suggestions:
In line 91 what does "rupture of the ventricular wall" mean?
In line 317, how can microcephaly missed on ultrasound be diagnosed on MRI, while the standards and definitions are based on ultrasound.
In line 320, what ultrasound diagnosis was rejected by MRI?
In line 413, why give a reference for you own data?
In Figure 1c, since there is also some germinal matrix hemorrhage, how was the presence of a choroid plexus papilloma confirmed and differentiated from a simple bleeding? was there hypervascularity in doppler ultrasound? Was there a post-delivery pathology or imaging study?
Now the general comments:
The paragraphs are long and difficult to read and not centered on a single main concept. There are sentences and fragments that are redundant and can be cut out to make the article easier and more pleasant to read. For example the sentence on lines 401-403 is completely circumstantial and not directly related to the paragraph purpose.
The conclusion is not what you aimed at as you state it in the introduction part. "
| we aim to bring new insights of the patterns of growth of the lateral ventricles in | 71 |
| regard of the brain parenchyma in isolated and non-isolated forms of ventriculomegaly." |
Overall, I find the data valuable, but presenting it in a more organized and clear way would greatly benefit the manuscript.
Thank you
Author Response
Reviewer Notes: “In the article you have shown the compatibility of the results in the two modalities; in addition you have shown differences in patterns of brain growth in isolated and non-isolated ventriculomegaly, concluding a different growth trajectories for the two groups.While such comparisons have been done before, the fact that that the fetal brains with additional structural abnormalities show wider divergence in growth in late pregnancy comparing to the cases of isolated ventriculomegaly is a significant finding. However while the isolated ventriculomegaly group is a relatively uniform group, I am not sure calling this divergence a growth of non-isolated ventriculomegaly a growth trajectory is proper in a very heterogeneous group.
We are grateful to the Reviewer for the time and effort and for their thorough review. We have made the requested changes and have addressed all the comments. We appreciate the thoughtful comments and have made all the requested revisions to the manuscript. Please find the details in the added information further below:
I also have some isolated suggestions, and find some issues in the overall presentation of the material. First the isolated suggestions:
In line 91 what does "rupture of the ventricular wall" mean?
Thank you for your question. Perhaps we needed to be more precise in explaining that we mean by “ruptured wall of the lateral ventricle” or the corrected term that we used for brevity - “ventricular rupture”. We found this terminology widely used in the medical literature. For example this term is used in a recent study by Beth M. Kline‐Fath et al. (from the Department of Radiology and Medical Imaging, Cincinnati Children's Hospital Medical Center, Cincinnati, OH, USA), published in 2018 in Prenatal Diagnosis (Prenatal Diagnosis. 2018;38:706–712). https://obgyn.onlinelibrary.wiley.com/doi/10.1002/pd.5317
We corrected our omission, and this could be found in line 172 of the manuscript!
In line 317, how can microcephaly missed on ultrasound be diagnosed on MRI, while the standards and definitions are based on ultrasound.
The Reviewer is correct – the microcephaly was indeed diagnosed by US. We agree that the sentence (line 994) does not clarify that the microcephaly had been diagnosed by US examination already. We have added in the Materials and Methods that our study group consisted of 40 cases in total, 12 of which had a primary diagnosis made by referral sonographers (before an expert neurosonographic follow-up) and 28 cases directly diagnosed initially at the routine visit by a clinician who had obtained The Fetal Medicine Foundation Certificate in neurosonography examination for fetal defects. We corrected the text to reflect this and this is now visible in lines 184-186 and 404-405. In the sub-section: Detection of additional anomalies on US and MRI we have added that we have compared the diagnosis of the referral sonographers with the diagnosis of the experts in neurosonography and finally with the MRI diagnosis in all cases. Microcephaly therefore was diagnosed on referral ultrasonography as the reviewer correctly suggested. Correction is done in lines 996-998 of the article.
In line 320, what ultrasound diagnosis was rejected by MRI?
In fact, it this concrete case, a suspicion of partial agenesis of corpus callosum was raised at the neurosonographic examination. This referral diagnosis was “not in agreement” rather than “rejected” by MRI. We agree with this valuable precise statement by the Reviewer and the correction is done in the manuscript in line 998-999.
In line 413, why give a reference for you own data?
Thank you for the correction of this error, it is an unintentional one and it has been corrected. We agree completely and apologize for the mistake with this reference. We did not mean to put a reference to our own data. The reference number 26 was inserted in the wrong place and should refer to a study, performed by Scott et al., described three sentences before, in line 1262. This has been corrected now.
In Figure 1c, since there is also some germinal matrix hemorrhage, how was the presence of a choroid plexus papilloma confirmed and differentiated from a simple bleeding? was there hypervascularity in doppler ultrasound? Was there a post-delivery pathology or imaging study?
The selected image in Figure 1c was chosen as an illustration of a case of severe unilateral ventriculomegaly, as Figure 1 illustrates the various degrees of ventriculomegaly. However, we selected this image as an example of an additional pathology, as we present in our study also non-isolated cases of VM. Thus, Figure 1c does not aim to focus on the pathology of the choroid plexus, which was interpreted as a hemorrhage from a papilloma of choroid plexus. This finding was confirmed by prenatal US follow-up, postnatal transfontanelle neurosonography and also by a postnatal contrast enhanced brain MR follow-up 10 months after delivery. Both studies observed an important reduction of the size of the hemorrhagic transformation within the plexus. No hyperechogenicities in the periventricular brain parenchyma and no hypervascularity on the transfontanelle cranial Doppler sonography were noted. We neither observed presence of a small germinal matrix hemorrhage on fetal nor on postnatal follow up MRI, based on prenatal SS-TSE T2 and T1 sequences and postnatal additional Flair, 3D Flair, SWi, DWI and ADC sequences. We would be more than happy to provide additional images to the Reviewer if required, both from this particular case or even the entire fetal MR study.
If the Reviewer considers the presented example of a severe unilateral VM unsuitable, we can replace the image in Figure 1c with another example of an isolated severe ventriculomegaly from our database.
Now the general comments:
The paragraphs are long and difficult to read and not centered on a single main concept. There are sentences and fragments that are redundant and can be cut out to make the article easier and more pleasant to read. For example the sentence on lines 401-403 is completely circumstantial and not directly related to the paragraph purpose.
With respect to the English Language, we agree and the manuscript has been through an extensive English language revision by a native English-speaking colleague with a substantial experience of publishing numerous articles. We are confident that the typographic, linguistic and grammatical errors have now been diligently corrected. This can be seen by the numerous Track Changes throughout the word document with respect correcting the English.
The conclusion is not what you aimed at as you state it in the introduction part.
Yes, we realize this now and we added two leading sentences to the conclusion as a confirmatory statement of the idea from the introduction.
Overall, I find the data valuable, but presenting it in a more organized and clear way would greatly benefit the manuscript."
We are grateful to see that the Reviewer considers that the study has generated valuable data. We hope that the revisited version of the manuscript will be easier to read after we addressed the English language and amended the manuscript with the helpful suggestions from the Reviewer. Thank you for the thorough and thoughtful comments and suggestions.
Reviewer 5 Report
Review Diagnosis
Esteemed Editor and Author Team,
I highly appreciate the manuscript and fully support its publication. Prenatal neurosonography with the adjunct of MRI are a powerful toolset for fetal assessment. It is mandatory to make continuous efforts to describe landmarks, create standardized planes and measurement algorhythms, provide gestational age and population adjusted growth charts for various developing fetal structures. This article represents one step in the right direction and, of course, it is not without certain limitations, as the authors have described. Further insight is needed to understand the clinical implications of imaging knowledge.
Author Response
We are pleased the Reviewer highly appreciates the manuscript and fully supports its publication. We are also pleased the Reviewer is satisfied that the manuscript presents clearly outlined methods and results, and a balanced discussion.
Round 2
Reviewer 4 Report
The manuscript looks better in this version. I still don't understand how can MRI make a diagnosis of microcephaly that ultrasound didn't. By definition microcephaly is defined by HC in ultrasound and measurements in US are much more established.
Author Response
We are very glad to see that the Reviewer considers the manuscript improved in this version and we are grateful for the comment addressed by the Reviewer!
We have included a specific sentence saying that microcephaly was diagnosed by an US examination.
To clarify it further, we have never highlighted in our manuscript that microcephaly was diagnosed by MRI first, however if this is the impression, then we have done the specific changes in order to avoid misunderstanding from the readers.
Thank you.